# OpenReview forum: "ExpeAttack: Efficient and Diverse LLM Jailbreaking via Two-Stage Exploration with Experience Replay"
_ICLR.cc/2026/Conference — Submitted to ICLR 2026_

### Official Review · Reviewer_EMXr · 2025-10-28

**Soundness:** 2
**Presentation:** 2
**Contribution:** 2
**Rating:** 4
**Confidence:** 4

**Summary:**

ExpeAttack is a framework for LLM jailbreaking that operates through a two-stage exploration with experience replay. The first stage, seed generation, employs a Pattern Factory to create diverse initial adversarial prompts by integrating various attack strategies like role-playing and semantic inversion.

The second stage, iterative refinement, utilizes a memory mechanism with short-term and long-term memory pools, along with an Insight-based Memory Compression mechanism, to distill successful attack patterns into transferable meta-instructions. This process guides the efficient and interpretable refinement of attack samples by treating jailbreaking as a task-solving problem where an adversarial agent autonomously explores and optimizes bypass strategies based on past experiences.

For evaluation, the method uses the Attack Success Rate (ASR), Average Number of Queries (ANQ), and a fine-tuning-based diversity evaluation against 100 harmful instructions sampled from the AdvBench dataset. It is benchmarked against the white-box baseline GCG and black-box baselines GPTFuzzer and PAIR.

**Strengths:**

The paper have the following two strengths:

- The Insight-based Memory Compression is novel. This mechanism is a creative theoretical combination that strategically distills successful attack histories into transferable meta-instructions, fundamentally redefining LLM jailbreaking as an experience-driven task-solving problem.

- The paper is well written. The paper clearly separates the process into seed generation via a Pattern Factory and iterative refinement guided by the memory module, making the complex dynamic optimization process highly digestible and interpretable.

**Weaknesses:**

The ExpeAttack paper exhibits three weakness that limit its scientific contribution.

1. the experimental evaluation is incomplete. The paper tests only 100 samples from AdvBench, while recent studies routinely evaluate across HarmBench, JailbreakBench, DAN, STRONGReject to demonstrate robustness. Single-benchmark evaluation fails to establish whether the method generalizes across diverse harmful content categories and different evaluation criteria.

2. the paper lacks reproducibility due to unspecified model versions. The authors list "Claude" as a target model without specifying which version was tested. Claude has multiple versions with substantially different safety alignments. Without precise model identifiers, the results cannot be reproduced or meaningfully compared with other research.

3. the paper provides no ablation studies to validate its claimed contributions. The authors claim three innovations but they provide no evidence demonstrating how each component independently contributes to attack success rates. Established research standards require detailed ablation analyses, as demonstrated by GPTFuzzer's studies.

4. The authors should demonstrate whether their Pattern Factory outperforms existing seed generation methods and whether combining their components with baseline methods like GPTFuzzer or PAIR yields additional improvements.


[1] Mazeika M, Phan L, Yin X, et al. Harmbench: A standardized evaluation framework for automated red teaming and robust refusal[J]. arXiv preprint arXiv:2402.04249, 2024.

[2] Chao P, Debenedetti E, Robey A, et al. Jailbreakbench: An open robustness benchmark for jailbreaking large language models[J]. Advances in Neural Information Processing Systems, 2024, 37: 55005-55029.

[3] Yu J, Lin X, Yu Z, et al. Gptfuzzer: Red teaming large language models with auto-generated jailbreak prompts[J]. arXiv preprint arXiv:2309.10253, 2023.

[4] Shen, Xinyue, et al. "" do anything now": Characterizing and evaluating in-the-wild jailbreak prompts on large language models." Proceedings of the 2024 on ACM SIGSAC Conference on Computer and Communications Security. 2024.

[5] Souly, Alexandra, et al. "A strongreject for empty jailbreaks." Advances in Neural Information Processing Systems 37 (2024): 125416-125440.

**Questions:**

I appreciate the novel approach presented in this work, but several critical aspects require clarification to properly assess the scientific contribution and validity of your findings.

1. Your evaluation uses only 100 samples from AdvBench, which raises concerns about generalizability. Could you address the following:

- What was the rationale for limiting evaluation to this single benchmark?
- Can you provide results on at least two additional standardized benchmark to demonstrate your method's robustness across diverse harmful content categories?

2. The paper lists "Claude" and "GPT-4" without version specifications, creating significant reproducibility gaps. Please clarify:

- Which specific Claude version was tested?
- Which GPT-4 variant was used?

3. Your paper claims three innovations but lacks ablation analysis. Could you provide individual performance metrics with each component removed independently?


4. To contextualize your contributions more clearly, please address:

- Does integrating your experience-based mutation strategy with existing frameworks like GPTFuzzer yield incremental improvements?
- Have you tested hybrid combinations to determine whether your components enhance existing methods?

---

### Official Review · Reviewer_eiDy · 2025-10-31

**Soundness:** 1
**Presentation:** 2
**Contribution:** 2
**Rating:** 2
**Confidence:** 4

**Summary:**

This paper introduces ExpeAttack, an experience driven method to continually generate and refine diverse jailbreaking strategies on closed source LLMs. To do this, the method uses a Pattern Factory to generate varied initial prompts through techniques like role-playing and semantic inversion, and employs long and short term memory to iteratively refine these prompts.

**Strengths:**

The main strengths of this paper are as follows:
1) The paper addresses a timely problem in stress-testing LLMs in the black box setting without access to gradients, logits, or other confidence measures.
2) The experiential learning strategy of iterative refinement using memory is novel.
3) The results show improved performance over the baseline models.

**Weaknesses:**

The main weakness of this paper is the experimental setup. My concerns are listed below:
1) The baselines used in this paper are not state-of-the-art anymore, or even close to it. GCG was developed in 2023, and since then there have been many more prominent methods in the white box setting: AutoDAN-HGA [1] , AutoDAN [2], BEAST [3], etc. Additionally in the black box setting there is: AutoDAN-Turbo [4], and MouseTrap [5] (for reasoning models). This also needs to be updated in the related work section, which seems to be limited to 2024?
2) The  LLMs used for jailbreaking are also not close to SOTA either, with Vicuna and Llama 2 in the open-source, and GPT-4o, GPT3.5, and Claude for the closed source LLMs. These are not safety aligned very well compared to existing models, so black box attacks will be more successful against these models. The paper needs to evaluate against current safety-finetuned models and reasoning models that could detect such strategies and navigate around them.
3) There is no ablation study for the components of the attack. The method is quite complicated, with different memory protocols, knowledge distillation, and sampling strategies. Ablation studies on each of the components, even limited to one dataset or one model, would greatly improve the understanding of this paper.

[1] Liu, X., Xu, N., Chen, M., & Xiao, C. (2023). Autodan: Generating stealthy jailbreak prompts on aligned large language models. arXiv preprint arXiv:2310.04451.

[2] Zhu, S., Zhang, R., An, B., Wu, G., Barrow, J., Wang, Z., ... & Sun, T. (2023). Autodan: interpretable gradient-based adversarial attacks on large language models. arXiv preprint arXiv:2310.15140.

[3] Sadasivan, V. S., Saha, S., Sriramanan, G., Kattakinda, P., Chegini, A., & Feizi, S. (2024). Fast adversarial attacks on language models in one gpu minute. arXiv preprint arXiv:2402.15570.

[4] Liu, X., Li, P., Suh, E., Vorobeychik, Y., Mao, Z., Jha, S., ... & Xiao, C. (2024). Autodan-turbo: A lifelong agent for strategy self-exploration to jailbreak llms. arXiv preprint arXiv:2410.05295.

[5] Yao, Y., Tong, X., Wang, R., Wang, Y., Li, L., Liu, L., ... & Wang, Y. (2025). A mousetrap: Fooling large reasoning models for jailbreak with chain of iterative chaos. arXiv preprint arXiv:2502.15806.

**Questions:**

1) A closed source LLM like GPT-4o is used for evaluating the attack strategy using a scale from 1-5. Is there any improvement in the performance of this method if a safety-aware reward model is used to do the evaluation?
2) Why is the performance of the method lower for GPT3.5 compared to GCG and GPTFuzzer?
3) Can you provide some successful and unsuccessful examples of the method working on current LLMs?
4) The ANQ provided is for queries to the victim model, but what about number of refinement steps, or FLOPs for knowledge distillation time?

**Details Of Ethics Concerns:**

There is no ethics statement on the security and safety issues for malicious actors to use the paper's method to jailbreak closed source LLMs. Please add such an ethics statement.

---

### Official Review · Reviewer_8Cmg · 2025-10-31

**Soundness:** 2
**Presentation:** 2
**Contribution:** 1
**Rating:** 2
**Confidence:** 3

**Summary:**

This paper introduces ExpeAttack, a novel framework designed to generate more efficient and diverse jailbreaking prompts for LLMs. ExpeAttack operates in two stages: Seed Generation: It uses a "Pattern Factory" to create diverse initial attack prompts based on various strategies like role-playing. Iterative Refinement: The system uses short-term and long-term memory pools. A key innovation is an "insight-based memory compression" mechanism that distills successful attack patterns into transferable "meta-instructions," which guide the refinement process efficiently. Experiments conducted across multiple LLMs demonstrate that ExpeAttack achieves higher ASRs compared with GCG, PAIR, and GPTFuzzer.

**Strengths:**

● The paper tackles the relevant and important problem of LLM jailbreaking, which is a significant and ongoing concern for the AI safety and security community.

● The authors propose a complete, end-to-end framework, ExpeAttack, which systematically organizes the attack process into distinct stages of seed generation and iterative refinement.

● The authors consider multiple evaluation metrics beyond simple attack success rate, including the efficiency (Average Number of Queries) and the diversity of the generated attacks.

**Weaknesses:**

● The novelty of this work is limited. The "Pattern Factory" for seed generation and the memory mechanism for refinement are existing in many works in this field, albeit in slightly different forms.

● The paper lacks comparison against more recent and relevant related work. The chosen baselines (GCG, PAIR, and GPTFuzzer) represent relatively early efforts in the field of automated jailbreaking.

● Figure 1 is not informative. Figure 2 is weakly linked to main text. For example, concepts mentioned frequently in main text (such as "pattern factory") do not appear in the figure, whereas terms appearing in the figure (such as "stratified sampling") are never explained in the main text.

**Questions:**

N/A

---

### Official Review · Reviewer_keov · 2025-11-01

**Soundness:** 2
**Presentation:** 3
**Contribution:** 2
**Rating:** 2
**Confidence:** 4

**Summary:**

The paper discusses the idea of storing long-term strategies (patterns), so that at test-time we can randomly sample from them are adapt new ones each time. It also discusses the idea of adaptive refinement of jailbreaking prompts through reflection and refinement. While both of these two ideas are explored in previous work, the combination of them is novel.

**Strengths:**

1- The paper discusses both long-term patterns (for better query efficiency and exploitation at test time) and adaptive refinement and provides a proper framework to combine them.

2- The authors have explained all elements of their method pretty well and establish the need for them. I believe that this direction is critical for jailbreaking methods and finding the vulnerabilities of models.

**Weaknesses:**

1- My main concern is that the authors have neither cited nor compared with the two main papers that have already studied long-term patterns and adaptive search via refinement. Can they justify this?

Lie et al., "AutoDAN-Turbo: A Lifelong Agent for Strategy Self-Exploration to Jailbreak LLMs"

Sabbaghi et al., "Adversarial Reasoning at Jailbreaking Time".

2- The authors have resorted to compare with methods that are not considered SOTA anymore even though they are quire established. Moreover, they only compare the baselines on obsolete models such as Llama-2 or Vicuna-7B. The comparison on Vicuna does not give any information since many methods achieve 100% ASR:

Andriushchenko et al., "Jailbreaking Leading Safety-Aligned LLMs with Simple Adaptive Attacks".

3- I found the diversity argument and the comparison with PAIR insightful. However, there are many algorithms that aim to generate diverse jailbreakings and again are not mentioned in the paper. For instance:

Samvalyan et al., "Rainbow Teaming: Open-Ended Generation of Diverse Adversarial Prompts".

**Questions:**

Please address the concerns above.

---

### Meta-Review · Area_Chair_6wx3 · 2026-01-06

**Summary:**

The paper discusses the idea of storing long-term strategies (patterns), so that at test-time we can randomly sample from them are adapt new ones each time. It also discusses the idea of adaptive refinement of jailbreaking prompts through reflection and refinement. While both of these two ideas are explored in previous work, the combination of them is novel.

All reviews gave negative scores and point out several severe weakness. At the same time, authors do not provide any rebuttals.
Thus, I recommend this paper should be rejected.

**Reviewer Scores:**

No

---

### Decision · Program_Chairs · 2026-01-26

Reject